# Tn5 Transposase Applied in Genomics Research

**DOI:** 10.3390/ijms21218329

**Published:** 2020-11-06

**Authors:** Niannian Li, Kairang Jin, Yanmin Bai, Haifeng Fu, Lin Liu, Bin Liu

**Affiliations:** 1College of Life Sciences, Nankai University, Tianjin 300071, China; liniannian123liu@163.com (N.L.); 2120181084@mail.nankai.edu.cn (K.J.); haifeng-fu@mail.nankai.edu.cn (H.F.); 2State Key Laboratory of Silkworm Genome Biology, College of Biotechnology, Southwest University, Chongqing 400700, China; baibai666@email.swu.edu.cn; 3School of Life Sciences, Tsinghua-Peking Joint Center for Life Sciences, Beijing Advanced Innovation Center for Structural Biology, Beijing Frontier Research Center for Biological Structure, Tsinghua University, Beijing 100084, China; 4TEDA Institute of Biological Sciences and Biotechnology, Nankai University, Tianjin 300071, China

**Keywords:** Tn5, 3D genome structures, genomic variation, open chromatin, long fragments, epigenetics

## Abstract

The development of high-throughput sequencing (next-generation sequencing technology (NGS)) and the continuous increase in experimental throughput require the upstream sample processing steps of NGS to be as simple as possible to improve the efficiency of the entire NGS process. The transposition system has fast “cut and paste” and “copy and paste” functions, and has been innovatively applied to the NGS field. For example, the Assay for Transposase-Accessible Chromatin with high throughput sequencing (ATAC-Seq) uses high-throughput sequencing to detect chromatin regions accessible by Tn5 transposase. Linear Amplification via Transposon Insertion (LIANTI) uses Tn5 transposase for linear amplification, haploid typing, and structural variation detection. Not only is it efficient and simple, it effectively shortens the time for NGS sample library construction, realizes large-scale and rapid sequencing, improves sequencing resolution, and can be flexibly modified for more technological innovation.

## 1. Tn5 Transposition Mechanism

Transposons are genetic elements that can “jump” to different locations within a genome. The first transposon was discovered in corn (maize) by Barbara McClintock [1]. Bacterial transposons can be divided into the following categories: Insertion sequences, Composite transposons, TnA family, and Muphage [2,3]. Tn5 is a compound transposon. Tn5 transposons were discovered in *Escherichia cdi* and consist of a core sequence encoding three antibiotics (neomycin, bleomycin, and streptomycin) and two inverted IS50 sequences, IS50L and IS50R, which encode a Tn5 transposase (Tnp) (Figure 1A) [2]. IS50 has two pairs of 19-bp inverted ends that are outside ends (OEs) and inside ends (IEs). These inverted OEs are target sites of Tn5 transposase [4]. When transposition occurs, transposases bind to the OEs of the Tn5 transposon, forming Tnp-OE complexes [5,6], and then the two complexes join together. The C-terminus of Tnp interacts and dimerizes [4] to form a synaptic complex that has the ability to cleave DNA [6,7]. Tnps that bind to the right and left ends are responsible for catalyzing the hydrolysis of the phosphodiester bond at the left and right ends, respectively [8,9]. Tnp activates water molecules that hydrolyze the DNA strand and forms a 3′-OH nucleophilic group at the 5′ ends of transposons, which in turn attacks the complementary strand to form a hairpin structure [8,9], that further forms a blunt-end by another activated water molecule. Finally, the synaptic complex captures target DNA and finishes the strand transfer by nucleophilic attack on both strands of the target DNA with 3’ OH groups of the Tn5 transposon (Figure 1B) [10].

The “cut and paste” function of Tn5 is widely used in genomics research. Subsequently, studies have shown that only OE sequences and Tn5 transposases are required for transposition in vitro [12]. Tn5 transposases can randomly insert adaptors/barcodes into DNA, and the resulting DNA molecules are ready for PCR amplification and sequencing [6,13]. The in vitro transposition elements of Tn5 for NGS library construction include the terminal sequence of the transposon, target DNA, transposase (Tnp), and Mg^2+^ (activator) [14,15]. The library structure formed by the transposition method is as follows: Tnp recognizes the end of the transposon to form a Tn5 transposition complex. The transposon sequence is conjugated with P5 and P7 end partial adapter sequence (Adapter 1/2) to form the donor DNA [16]. The complex recognizes the target sequence of the acceptor DNA, cuts the acceptor DNA, and inserts the carried donor DNA to form DNA with a P5 part adapter Adapter 1 at one end and a P7 part adapter Adapter 2 at the other end, which is then added by PCR barcoding, and the rest of the linker form a DNA library with complete linkers at the P5 and P7 ends (Figure 2A) [17].

Based on these principles, Tn5 transposase, including more sensitive versions [18], has been widely used in many fields, e.g., determination of the three-dimensional (3D) structure of human cells (Dip-C) (the enlarged one has added as [22]] (Figure 2D in this [22]] (Figure 1C footer) [17]. An analysis of functionally annotated genic regions revealed a sharp decrease in CpG methylation (Tn5mC-seq) [19], introducing biotin in the process of long-range sequencing (Mate-pair) [20], visualization of accessible chromatin (Assay for Transposase-Accessible Chromatin with high throughput sequencing (ATAC-Seq)) [21] and single-cell SNP detection (Linear Amplification via Transposon Insertion (LIANTI)) [22] (Figure 1C). Here, we review the principles of Tn5-based techniques and their applications [23], ranging from embryonic development to human cancer [23]. We also focus on cutting-edge Tn5-based techniques that can be used in other fields.

## 2. Application of Tn5 to 3D Genome Structures

The genomic structure of mammalian cells is complex and consists of DNA, histones, and other regulators that control gene expression, cell fate, and other cellular functions [24]. Tn5 transposition technology has been widely used in high-throughput sequencing for the study of chromatin 3D structure. Chromatin conformation capture assays such as 3C [25] and Hi-C [26] allow reconstruction of 3D genomic structures using data on genome-wide neighboring loci from bulk samples. These technologies all include the application of Tn5 in high-throughput sequencing. However, characterization of the 3D genomic structure of single mammalian cells using these methods are challenging to due to the loss of a large amount of DNA sequence information [17]. Recently, a method called Dip-C was developed to reduce the loss of DNA information and reshape the 3D genomic structure of diploid mammalian cells using the Tn5 system and 20 barcodes in sequencing library construction [17], instead of only the two barcodes that are typically used in the system [17]. 

Of course, these techniques for chromatin 3D structure research, including 3C, Hi-C and Dip-C, apply the same basic principles of Tn5 transposition. To accomplish DNA interruption, the P5 and P7 terminal partial adapters (Adapter 1/2) and the transposon end sequences are used to form a coating adapter to further assemble a Tn5 transposable complex with Tnp [27,28]. This complex interrupts the receptor DNA and forms DNA with a P5 partial adapter (Adapter 1) at one end and a P7 partial adapter (Adapter 2) at the other end. Then, the barcodes are added by PCR to form a sequencing library [29,30]. In another approach, the barcode and the transposon end sequence form a coating adapter and further form a Tn5 transposon complex with Tnp. Once transposition is complete, the adapter is added by PCR (Figure 2A) [31]. However, in the actual operation process, when we use the Nextera kit (Illumina) to construct sequencing libraries, two different barcodes are randomly inserted into the incoming DNA. These two barcodes then act as two PCR primers to amplify the resulting fragments. Reads with two different barcodes at both ends of each read are enlarged, while reads with two identical tags are lost. As a result, at least 50% of the input DNA is lost after PCR amplification (Figure 2B) [17]. The reduction in the amount of information directly affects the resolution for simulating the 3D genome structure. In multiplex end-tagging amplification (META) (Xing et al., US provisional patent 62/509,981), the loss of input DNA is markedly reduced by inserting N different barcodes. Thus, only 1/n of the initial DNA is lost. Illumina adaptors are later added using two short PCR steps (Figure 2C) [17]. Using META for DNA amplification, a new method called Dip-C has been developed to reconstruct the genome structure of a single diploid human cell from a lymphoblast cell line and primary blood cells at 1-kb resolution (Figure 2D) [17], while the resolution is 25-kb for bulk Hi-C.

## 3. Application of Tn5 to Study Genomic Variation

Single-cell genomics is important for biology and medicine. However, current methods for whole-genome amplification (WGA) are limited by the low accuracy in detecting copy number variations (CNVs) and the low fidelity of amplicons [22]. These technologies are based on the characteristics of random exponential amplification. Exponential amplification will lead to unequal amplification (the difference in magnification by area varies greatly), and exponential amplification is precisely the single-cell genome amplification technology. However, this seems to be a natural paradox because when enough DNA is obtained, genomic DNA must be randomly exponentially amplified. The transformation of the Tn5 structure with LIANTI technology cleverly solved this problem because the entire genome amplification process uses linear amplification technology. LIANTI uses linear amplification to obtain a sufficient amount of DNA and see the basic decomposition: First, Tn5 transposase was used to combine the LIANTI sequence (green part is T7 promoter), and then the Tn5 transposase complex was used to randomly insert single-cell genomic DNA. The insert contains a T7 promoter, and subsequent transcription is used to obtain a large number of linearly amplified transcripts. Then, after reverse transcription, numerous amplification products are obtained. The entire amplification process uses a linearized transcription process without any exponential amplification steps. Here, a detailed comparison is given between exponential and linear magnification (Figure 3B) [22]. It is assumed that DNA fragments A and B have replication yields of 100% and 70% per round. For a final copy number of about 10,000, the ratio of fragment A/B for exponential amplification is 8:1, which hinders the accuracy of CNV detection. In contrast, that ratio using linear amplification is much smaller (1:0.7, Figure 3A,B) [22]. Linear amplification is also superior to exponential amplification in terms of fidelity. In exponential amplification, the highest-fidelity polymerase that replicates the human genome (3 × 10^9^ bp) in the first cycle generates about 300 errors that will propagate permanently in the next replication cycle, leading to false positive SNVs [22]. In contrast, using linear amplification, errors randomly appear at different positions of the amplicon and can be easily corrected (Figure 3A,B) [22].

To construct a LIANTI sequencing library, several steps are performed (Figure 3C) [22]. First, equimolar amounts of LIANTI transposon and Tn5 transposase are mixed and dimerized to form a LIANTI transposome [22]. Second, the fragmented genomic DNA from a single cell is transcribed into RNA with the help of LIANTI transposons that contain T7 promoter sequences. Then, RNA is reverse transcribed into cDNA for further barcode addition and DNA sequencing. With this method, genome-wide replication origin firing and replicon formation can be detected based on the increase in the copy number in 11 single cells with 10-kb bin size (approximately 250 Mb Chr1, as shown in Appendix A) [22]. Using LIANTI, the correlation between the density of UV-induced SNVs and the minus Rep-Seq signal reflecting the genomic region of replication, and DNase I hypersensitivity signal could also be detected (Appendix A) [22].

## 4. Tn5 Application in Open Chromatin

Chromatin has two states: euchromatin (accessible chromatin) and heterochromatin (non-accessible chromatin) [32]. Euchromatin has a low degree of compression, is stretched, and is more actively transcribed [33,34]. Heterochromatin has a high degree of folding and compression, is in a condensed state and has no transcriptional activity [35]. Beyond the invention of the ATAC-seq technology [36], there have been many methods to study open chromatin. For example, DNase-seq [37]. uses DNase to cut genomic DNA regions with free protein binding, and the trimmed complexes are isolated [37]. Then, the DNA from the trimmed complexes is sequenced and compared with known whole-genome sequences. Consequently, the regions with no read coverage are the open chromatin areas (Figure 4A) [38]. The disadvantages of this method are that it is time-consuming (2–3 days), laborious, and indirectly obtains sequences from open chromatin [38]. Furthermore, it requires a large number of cells, i.e., generally 10^6^ cells. In addition, repeatability is poor [38]. FAIRE-seq uses formaldehyde to fix the cells [39], and then uses ultrasonic waves to break the chromatin and isolates the interrupted DNA by chloroform extraction (Figure 4B) [39,40]. The steps are complicated, and the operation takes a long time. To overcome the shortcomings from both DNase-seq and FAIRE-seq, Tn5 transposases have been used to cut genomic DNA while adding adapters. Then, the PCR amplification products can be directly sequenced (Figure 4C) [36], Extensive analyses have shown that ATAC-seq provides accurate, direct, and sensitive measurements of chromatin accessibility [36]. However, these methods require a pool of cells, which indicates that the data collected reflect cumulative accessibility for all cells. Greenleaf and his colleagues integrated ATAC-seq into a programmable microfluidics platform for single-cell ATAC-seq (scATAC-seq) to uncover cell-to-cell regulatory variations in human cells [41].

Comparison of the results of DNase-seq, FAIREseq, and ATAC-seq revealed that the signal-to-noise ratio of the data from ATAC-seq is similar to that of DNase-seq, whereas FAIRE-seq has a lower value. The peak intensity is highly reproducible between technical repetitions and has a high correlation between ATAC-seq and DNase-seq (Appendix A) [36]. In summary, ATAC-seq [36] has the advantages of short time for DNA fragmentation, ease in DNA enrichment, and accurate amplification (Figure 4E). Although ATAC-seq is superior, there is room for improvement in the sensitivity of ATAC-seq. Using engineered Tn5 enzyme, which is more efficient and specific insertion of adaptors into open chromatin, Sos et al. have developed a new method called transposome hypersensitive site sequencing (THS-seq) to capture additional regulatory regions in bulk cells [42]. This further improvement in ATAC-seq and THS-seq could be applied in both fixed and unfixed samples [43].

In addition, Chen et al. have developed an assay involving transposase-accessible chromatin with visualization (ATAC-Seq) [21], in which Tn5 transposases insert fluorescent DNA adaptors into open chromatin loci, and the cells are imaged with super-resolution microscopy (Figure 4D) [21]. To apply ATAC-Seq, four-color imaging combining lamin B1, ATAC-Seq, DAPI, and mitochondrial protein markers was developed to clearly depict accessible DNA in the nucleus and reveal a strong overlap between extranuclear signals from the mitochondria and ATAC-Seq [21]. After visualization, the open regions in chromatin could also be investigated with ATAC-seq [21]. It has multiple potential applications; for example, human clinical specimens, FACS sorting, and clinical diagnosis.

## 5. The Application of Tn5 in Long Fragments

Long-fragment information is required for de novo assembly, structural variation detection, and haplotype phasing. How is information on long fragments obtained using the Tn5 complex? Without the presence of magnesium ions, transposome is stable, and multiple transposomes can be formed on a single long-range DNA molecule (Figure 5A) [30,44]. Based on this mechanism, Peters et al. have developed a technology called long fragment read (LFR) [44]. In LFR, the fragments are physically separated into 384-well plates, and a unique barcode for each well within transposomes is added. Then, samples from each well are merged and sequenced using second-generation sequencing. Finally, long fragment information is assembled using the unique barcodes (Figure 5B) [45]. Later, the single-tube LFR (stLFR) technology emerged and is regarded as a breakthrough technology that can complete all reactions in one tube, significantly reducing the complexity and time needed for long-fragment library construction (Figure 5C) [44]. In stLFR sequencing library construction, microbeads are used. The surface of each microbead carries a few unique barcode sequences that are transferred to the subfragments of each long DNA molecule by the Tn5 complex [46]. Generally, 10,000- to 100,000-bp single DNA molecules can be directly sequenced using third-generation sequencing technologies [47]. However, compared with second-generation sequencing, third-generation sequencing is relatively expensive, has low accuracy, and low throughput. Thus, in the future, Tn5 transposase-based long-fragment sequencing still has competitive strength in the sequencing market.

## 6. The Application of Tn5 in Epigenetics

Epigenetics refers to changes that influence gene expression that can be heritable through cell division and yet can be reversible without changes to the DNA sequence. DNA methylation is a broad epigenetic modification that plays a key role in genomic regulation [48]. Whole-genome bisulfite sequencing (WGBS) is the most comprehensive and high-resolution method for detecting DNA methylation [49,50]. However, a limitation of WGBS is that it requires more than 5 mg of input genomic DNA for each sample, which is impossible for most in vivo experiments [50]. Biologists have been relying on bisulfite sequencing to detect 5mC and 5hmC modifications for decades, but this chemical is extremely destructive, and 99% of DNA is degraded when it is exposed to it. Therefore, a large number of DNA samples are required. Adey et al. have described an improved method called Tn5mC-seq that reduces the amount of starting material by more than 100-fold relative to WGBS (Figure 6A) [19]. In Tn5mC-seq, all cytosine residues in the adaptors are methylated to maintain cytosine identity during bisulfite treatment [19]. Then, the unmethylated cytosines in the fragmented genomic DNA are converted into uracil using standard bisulfite treatment [19]. Finally, the methylation sites are detected by the comparison of sequences between reads and the reference genome [19]. Tn5mC-seq was used to detect the ratio of methylated cytosine in total cytosine in a 10-kb window on human chromosomes (Appendix A) [19] and the ratio of methylated CpG in total CpG residues at the annotated locus (Appendix A) [19]. Methylation is dynamic in the enhancer regions during cell fate transitions, but current models insufficiently define its role in gene regulation.

Hodges et al. have developed the ATAC-me technology, which inherits the advantages of ATAC-seq and can simultaneously detect DNA methylation and chromatin accessibility at enhancers of steady-state cells [51]. In ATAC-me, ATAC-seq is first used to add adaptors for DNA from open chromatin regions, and the resulting fragments go through a bisulfite treatment to mark methylation sites (Figure 6B) [51]. Using ATAC-me, a significant disconnect between chromatin accessibility, DNA methylation status, and gene activity is observed, which implies that it is important to construct precise molecular timelines to precisely understand the role of methylation in regulating gene expression [51].

The most direct evidence for the mechanisms of gene expression regulation and cell fate determination is the interaction of a specific chromatin region with proteins [52]. The chromatin immunoprecipitation with sequencing (ChIP-seq) and CUT&RUN [53] can achieve epigenomic profiling, but suffer from low signals, high background, and low yield due to the requirement of a large number of cells. To overcome these limitations, Kaya-Okur et al. have developed a method, Cleavage Under Targets and Tagmentation (CUT&Tag), which uses Protein A-fused Tn5 transposase (pA-Tn5) in transposomes to concurrently bind to the second antibody and tag the adapters into DNA around the proteins of interest [54]. Using CUT&Tag, the profiles for H3K4me1 and H3K4me2 histone modifications have been investigated in K562 cells. The subtle use of Tn5 is also reflected in ChIL–seq [55,56], a method for detecting genomic histone modifications in ultra-small number of cells, which use Tn5 to complete the transposition of ChIL probe that comprises the secondary antibody and ChIL DNA containing a T7 promoter and a primer sequence for the sequencing library preparation, including a mosaic end for Tn5 transposase binding [55,56]. An alternative method, the combinatorial indexing design (CoBATCH), has also been designed for single-cell epigenomic profiling (Figure 6C) [57]. CUT&Tag and CoBATCH both use specific antibodies and the pA-Tn5 transposases in adaptors’ tagmentation, and both have been utilized for single-cell epigenomics studies. There are differences between these two methods. Cut&Tag uses the SMARTer ICELL8 single-cell system to array single cells after the first- and second-antibody, and pA-Tn5 transposome incubation, whereas in CoBATCH, single-cell information is determined by bioinformatics analyses based on two rounds of barcode addition, transposition unique barcode, and, in addition, unique PCR index primers for each well of a 96-well plate (Appendix A) [57].

Later, He’s lab also reported a new method called simultaneous indexing and tagmentation-based ChIP-seq (itChIP) [57], which is used for histone modification and non-histone protein-binding profiling. itChIP starts with as few as 100 crosslinked cells. Then, it loses the chromosomes of cells with SDS under 62 °C, evenly inserts adaptors with Tn5 tagmentation and pulls the interested chromatin DNA-binding proteins with antibodies. Finally, PCR enrichment and sequencing are conducted (Figure 6D). By sorting single cells into each well of the 96-well plate using fluorescence-activated cell sorting (FACS), He’s lab has achieved single-cell itChIP and obtained ~9000 reads per cell to study the epigenetic route to exit from naive mouse ESCs and lineage-specific enhancer usage along cardiac progenitor cell fate determination [58]. It is distinct from ChIPmentation [52], in which adaptors for PCR amplification and sequencing are added by the Tn5 transposome to antibody-immunoprecipitated chromatin from the cell extraction, while itChIP inserts barcoded adaptors into nuclei fixed cells (Appendix A) [58]. In addition, to compare with the CUT&Tag method, itChIP has higher fraction of reads in peaks (FRiP) (52% vs. 27%) and sensitivity (21% vs. 6%). With minor modifications in Tn5 tagmentation protocols, other sequencing methods have been developed in epigenetics, e.g., antibody-guided chromatin tagmentation (ACT-seq) [53].

## 7. Challenges and Bottlenecks of Tn5-Related Techniques

The application of Tn5 transposases also has some challenges and bottlenecks. First, Tn5 transposases require DNA that is usually more than 1000 bp in length. Therefore, samples with short DNA fragments are not particularly suitable for constructing sequencing libraries, e.g., non-invasive prenatal examination (NIPT) and liquid biopsies that contain approximately 170-bp circulating free DNA [59]. Second, the ratio of DNA to Tn5 enzyme needs to be 1.5, and thus it requires high DNA quantification in preparation of DNA [22]. Third, some impurities in solution will affect the interruption effect of the Tn5 enzyme, and DNA-binding proteins inhibit the uniform insertion of adapters into DNA by the enzyme [60].

## 8. Conclusions and Perspectives

Tn5 transposases are versatile enzymes that randomly cut DNA and simultaneously insert transposons (adapters) into DNA, and the resulting fragments are ready for PCR amplification and sequencing. To date, based on its transposition feature, many methods have been developed to address various questions in the life sciences using fixed or unfixed samples from tissues, bulk cells, and single cells.

In the future, these Tn5 transposase-involved or -modified methods could be applied to other fields. In live cells, mRNAs are highly regulated, and its abundance partially reflects the function of genes. Recently, Cole et al. have introduced a method, Tn5 Prime, that defines the start sites of transcription and simultaneously estimates the expression levels of mRNAs of bulk cells or single cells [61]. Because transposition occurs on double-stranded DNA using Tn5 transposomes, the construction of RNA sequencing libraries is used for reverse transcription and second-strand synthesis before Tn5 tagmentation of the resulting dsDNA [62]. However, Di et al. found that Tn5 tagmentation can be done on RNA/DNA hybrids, and based on this principle, they built RNA-seq libraries within four hours (Appendix A) [63]. To use different barcodes in Tn5 transposome for each sample, this new RNA-seq method can be used to detect viruses, e.g., COVID-19, for tens of hundreds of clinical samples in an RNA-seq library. Genome editing with CRISPR-Cas9 is currently the hottest tool for the study of gene functions in many organisms. By combining the single-cell CRISPR screening, Rubin et al. introduced perturbation-indexed single-cell ATAC-seq (Perturb-ATAC), in which the chromatin accessibility that was controlled by transcription factors, chromatin modifiers, and noncoding RNAs was determined in GM12878 cells [64]. With the help of both ATAC-seq and CUT&Tag techniques, Dan et al. have revealed PAC1, a phosphatase, which suppresses T cells and attenuates host antitumor immunity [65]. Thus, it is a new direction to combine Tn5-related methods with other techniques to address new issues. Intracellular bacteria survive in host cells by secreting effectors to regulate gene expression [66]. SteE, an effector protein of *Salmonella*, has the ability to force mammalian serine/threonine kinase GSK3 to phosphorylate the non-canonical substrate signal transducer and activator of transcription 3 (STAT3) on tyrosine-705, which converts macrophages from the M1 to the M2 state [67]. However, how this phosphorylated STAT3 modulates chromosome accessibility and the expression of genes remains unclear. In the future, Tn5-related methods such as CoBATCH and single-cell itChIP could be used to uncover how bacteria effectors dynamically influence chromosomal states at the single-cell level.

Tn5 has also been extensively used to knock-in genes, create mutant libraries, study gene essentiality, and create reduced genomes [68,69]. To identify new virulence genes, a mutant library of *Xanthomonas citrus* subsp. The EZ-Tn5 transposon was used to produce citri 29-1, and the mutant was inoculated into susceptible grapefruit. Forty mutants with altered virulence phenotype were identified. The use of Tn5 transposons to detect the expression of these genes is essential for the development of ulcerative disease [70]. Tn5-related methods could be used in developmental biology because of the existence of epigenetic events during development [71]. For example, a whole X chromosome is “silenced” to transcribe mRNA in the early development of female embryos, which is called X-chromosome inactivation (XCI). A long non-coding RNA (lncRNA), Xist, and protein SPEN are believed to orchestrate this process [72]. Dossin et al. used CUT&RUN to map the location of SPEN on the X chromosome and revealed that SPEN associates with active gene promoters and enhancers shortly after the start of Xist expression and disengages from these sites after XCI. As stated earlier, Tn5 techniques have been extensively used in epigenetics and have outperformed the CUT&RUN technique. Therefore, Tn5 transposases may be utilized as powerful enzymes in developmental biology.

## Figures and Tables

**Figure 1 ijms-21-08329-f001:**
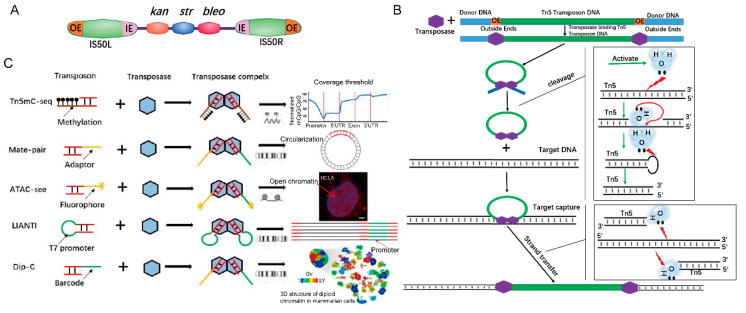
Tn5 transposon structure and transposition mechanism. (**A**). Tn5 transposon structure consisting of a core sequence that encodes three antibiotics and two inverted IS50 sequences. The outside ends (OEs) bind to Tn5 transposases. (**B**). Scheme of the Tn5 transposition mechanism [11]. (**C**). Tn5 adaptor modification is used in epigenetics, genomic structure, and chromatin visualization. Tn5mC-seq is used in research studies on DNA methylation. Mate-pair applied to the amplification of long DNA fragments. Assay for Transposase-Accessible Chromatin with high throughput sequencing (ATAC-Seq) is applied to open chromatin (scale bar 1:600). Linear Amplification via Transposon Insertion (LIANTI) is applied to genomic variation. Dip-C is used in reconstructing the 3D structure of the genome.

**Figure 2 ijms-21-08329-f002:**
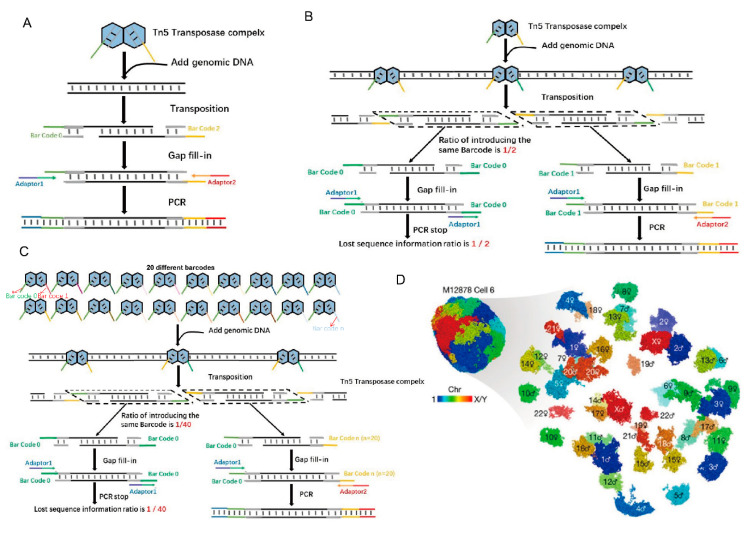
Tn5 in reconstructing 3D genome structures. (**A**). Tn5 is used to form a composite with barcodes, and then adapters are added by PCR. (**B**). The traditional method of Tn5 transposition using two barcodes results in the loss of 50% of the genomic sequence information. (**C**). Dip-C sequencing library construction utilizes 20 barcodes, reducing the initial information loss to 1/20 after PCR [17]. (**D**). Tan et al. used Dip-C to simulate the 3D genome structure of a human cell [17].

**Figure 3 ijms-21-08329-f003:**
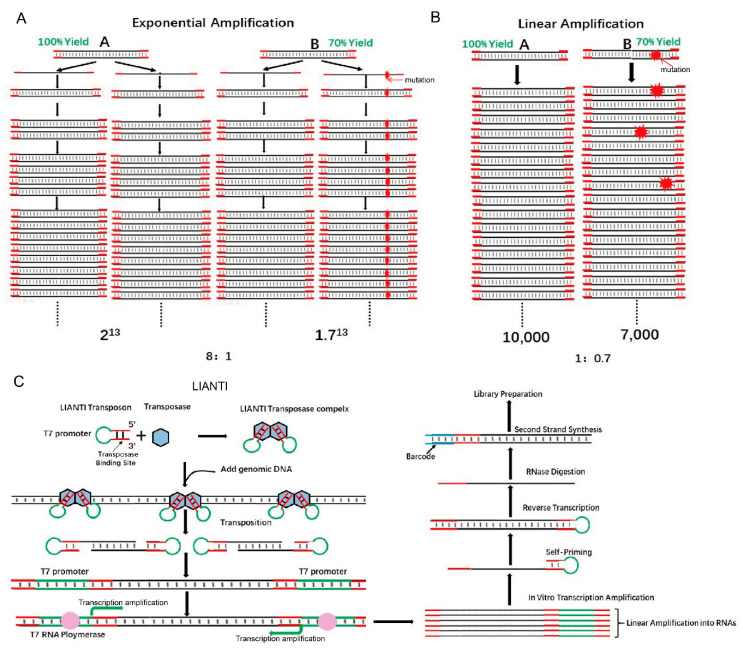
Tn5 is used in the study single-cell genomic variations. (**A**). Exponential amplification results in bias and errors. The replication yields of DNA fragments A and B are 100% and 70%, respectively. For a final copy number of approximately 10,000, the final ratio of fragments A/B for exponential amplification is 8:1 [22]. (**B**). Linear amplification significantly reduces bias and errors. The replication yields of DNA fragments A and B are 100% and 70%, respectively. For a final copy number of approximately 10,000, the final ratio of fragments A/B for linear amplification is 1:0.7 [22]. (**C**). The scheme of LIANTI sequencing library construction [22].

**Figure 4 ijms-21-08329-f004:**
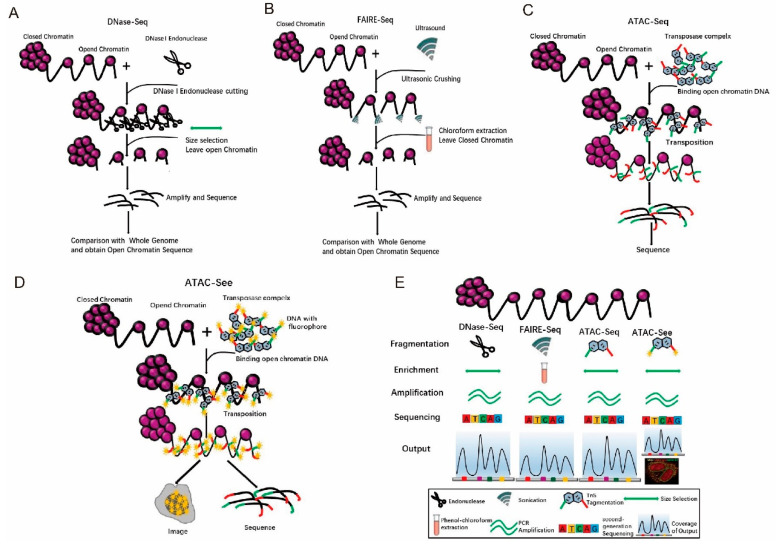
Tn5 is used to study open chromatin. (**A**). The principles and processes of DNase-seq. (**B**). The principles and processes of FAIRE-seq. (**C**). The principles and processes of ATAC-seq. (**D**). The principles and processes of ATAC-Seq. (**E**). Comparison of multiple methods for studying open chromatin.

**Figure 5 ijms-21-08329-f005:**
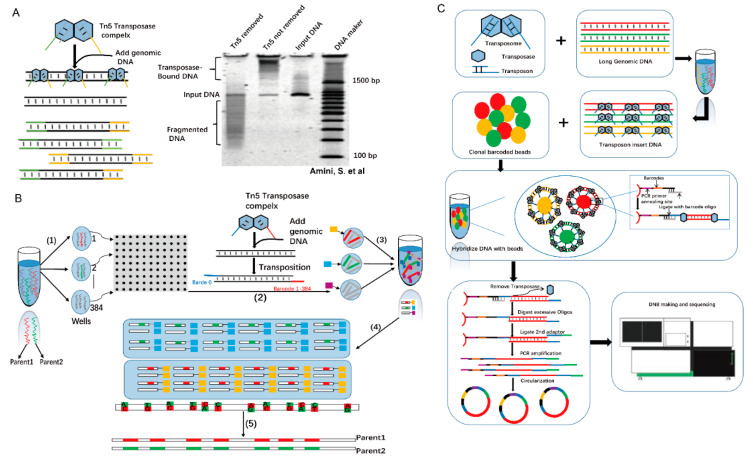
Tn5 is used in long-fragment sequencing. (**A**). PAGE analysis of transposase continuity. Tn5 transposons are used to target 1-kb PCR amplicons. Lane 1, treatment of transposome with SDS to remove transposase; Lane 2, transposome without SDS treatment is used as control. Lane 3, input DNA; Lane 4, a DNA marker. Tn5 transposase remains bound to DNA after transposition, and the protein-DNA complex dissociates only after the addition of the protein denaturant, SDS [30]. (**B**). Principle and scheme for LFR. (1) Physical separation of 100-130 pg of high-molecular weight DNA into 384 different wells. (2) Through several steps, all in the same well, without intermediate purification, genomic DNA is amplified, fragmented, and ligated onto a unique barcode adapter (3) that is merged to all 384 wells, purified, and introduced into Complete Genomics’ sequencing platform 10, (4) custom alignment program to map paired reads to the genome, and the barcode sequence is used to group tags into haplotypes. (5) The final result is the diploid genome sequence. (**C**) Schematic diagram of stLFR technology. This technique starts from the extracted long DNA and inserts the transposon sequence into the long DNA randomly. The DNA double-strand complementation principle is used to combine the product with a magnetic bead carrier with multiple copies of molecular tags. After two adapters, PCR amplification is performed, and library construction is finally completed.

**Figure 6 ijms-21-08329-f006:**
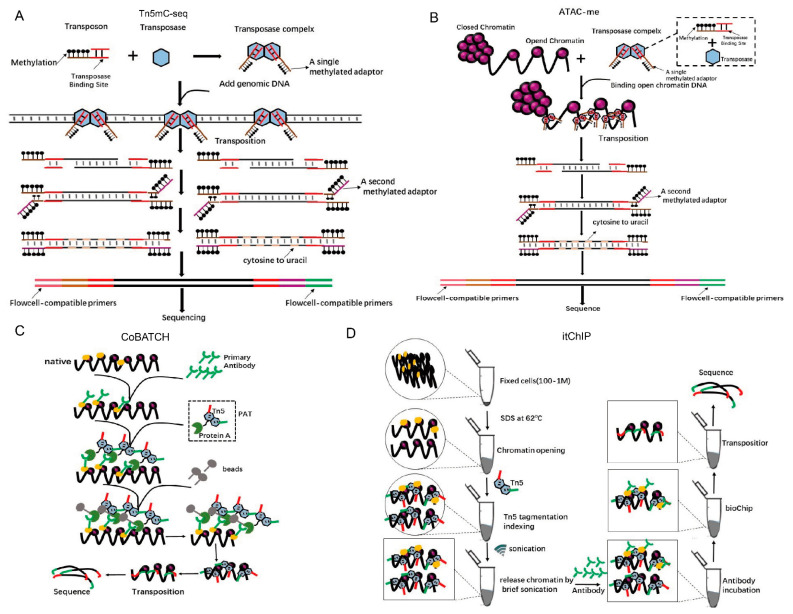
Using Tn5 in epigenetics. (**A**). Principle and s cheme for Tn5mC. Tn5 transposases loaded with a methylated adaptor (brown) attack genomic DNA. Oligonucleotide replacement methods anneal the second methylated adaptor (purple) and perform gap repair. Bisulfite treatment converts unmethylated cytosine to uracil (gray) and PCR is performed to add primers (pink, green) that are compatible with the external flow cell. Methylation is represented as a black lollipop. (**B**). Principle and scheme of ATAC-me. The experimental procedure is similar to Tm5C, but it is only for methylation of open chromatin. (**C**). A new technology called the combinatorial indexing design (CoBATCH). Protein A is fused to the N-terminus of a Tn5 transposase to form Protein A-Tn5 (PAT). First, antibodies are used to target a specific protein that binds with chromatin DNA. Then, PAT transposomes are used to insert the adaptors into antibody-immunoprecipitated chromatin and the resulting DNA is sequenced. (**D**). Principle and scheme of a new chip-seq technology called itChIP. Cross-link samples (cells or tissues) are treated with SDS at 62 °C to loosen whole-genome chromosomes without affecting the binding of proteins to DNA. Using this treatment, Tn5 can evenly cut chromosomes without creating a preference for open regions. Finally, antibodies are used to pull the specific proteins that bind to chromosomal DNA with adapters that are ready for PCR amplification and sequencing.

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
