# Peer review of "Tn5 Transposase Applied in Genomics Research"

_ijms, 2020, doi:10.3390/ijms21218329_

Round 1
Reviewer 1 Report
The manuscript of Li et al basically deals with technology implementing Tn5 transposition, reviewing protocols and applications.
Basically, the manuscript appears well-written and detailed. Though, it looks more like a technical report than a review, and it is not clear the aim of Authors. Manuscript is very detailed in describing protocols and application, but what it is really lacking is a review of practical examples of studies where Tn5 has been used (for different analyses). In this way, readers outside the field (more or less like me) can really appreciate the lab protocol but do not have a clear examples of what it can be expected in term of results and conclusions. The addition of such example may help to better understand the utility of Tn5 technique.
Moreover, it is not clear the context of Tn5 as a transposon: a brief overview of the evolution of Tn5, taxonomic distribution and so on (even limited to a few model organisms) would make the review more complete.
Author Response
Basically, the manuscript appears well-written and detailed. Though, it looks more like a technical report than a review, and it is not clear the aim of Authors. Manuscript is very detailed in describing protocols and application, but what it is really lacking is a review of practical examples of studies where Tn5 has been used (for different analyses). In this way, readers outside the field (more or less like me) can really appreciate the lab protocol but do not have a clear examples of what it can be expected in term of results and conclusions. The addition of such example may help to better understand the utility of Tn5 technique.
Response: Thank you very much for your comments and suggestions, which were of great help in improving our manuscript. According to your suggestions, we have added practical examples of studies where Tn5 has been used (for different analyses) in line 57-60.
Moreover, it is not clear the context of Tn5 as a transposon: a brief overview of the evolution of Tn5, taxonomic distribution and so on (even limited to a few model organisms) would make the review more complete.
Response: Thank you very much for your comments and suggestions. We added a brief overview about Tn5 in the first part of "Tn5 transposition mechanism".
Reviewer 2 Report
Comments for Authors:
The manuscript, submitted by Li et al., describes the mechanism of the Tn5 transposon and its application to various NGS-based technologies. Although tn5 is used by many researchers, the breadth of its use is so wide that many people are not fully aware of the areas in which tn5 is being used. This is a very good review and will be very useful to many researchers who used various NGS-related techniques.
I have a few points that the authors should address:
1) Abstract:
TN5 should be replaced by Tn5 to keep consistency in the manuscript.
2) Page 2, lines 49–55; " The library structure formed by the transposition method is as follows: ..."
This part may be difficult to understand for non-familiar readers without a figure. Please consider adding a figure to explain the section.
3) Page 3, line 106; "Tan et al. used Dip-C to simulate the 3D genome structure of a human cell."
The reference number (Ref.18?) may be provided here.
4) In addition to the ChIP-related techniques introduced in the manuscript, Harada et al. recently reported ChIL-seq, which is a ChIP-like epigenetic technology using Tn5 transposon and requires only a small number of cells. Please consider mentioning about this method in the manuscript.
Harada, A., Maehara, K., Handa, T., Arimura, Y., Nogami, J., Hayashi-Takanaka, Y., ... & Ohkawa, Y. (2019). A chromatin integration labelling method enables epigenomic profiling with lower input. Nature cell biology, 21(2), 287-296.
Handa, T., Harada, A., Maehara, K., Sato, S., Nakao, M., Goto, N., ... & Kimura, H. (2020). Chromatin integration labeling for mapping DNA-binding proteins and modifications with low input. Nature Protocols, 1-27.
Author Response
The manuscript, submitted by Li et al., describes the mechanism of the Tn5 transposon and its application to various NGS-based technologies. Although tn5 is used by many researchers, the breadth of its use is so wide that many people are not fully aware of the areas in which tn5 is being used. This is a very good review and will be very useful to many researchers who used various NGS-related techniques.
Response: Thank you very much for your comments and suggestions, which were of great help in improving our manuscript. We have revised the manuscript according to your comments and suggestions.
I have a few points that the authors should address:
1) Abstract:
TN5 should be replaced by Tn5 to keep consistency in the manuscript.
Response: Thank you very much for your suggestions. Done.
2) Page 2, lines 49–55; " The library structure formed by the transposition method is as follows: ..."
This part may be difficult to understand for non-familiar readers without a figure. Please consider adding a figure to explain the section.
Response: Thank you very much for your suggestions. We described the library structure formed of the Tn5 transposition in Figure 2A.
3) Page 3, line 106; "Tan et al. used Dip-C to simulate the 3D genome structure of a human cell."
The reference number (Ref.18?) may be provided here.
Response: Thank you very much for your suggestions. According to your suggestion, the reference number have been added.
4) In addition to the ChIP-related techniques introduced in the manuscript, Harada et al. recently reported ChIL-seq, which is a ChIP-like epigenetic technology using Tn5 transposon and requires only a small number of cells. Please consider mentioning about this method in the manuscript.
Response: Thank you very much for your comments and suggestions, which were of great help in improving our manuscript. We have introduced this method in the part of “The application of Tn5 in epigenetics”. The subtle use of Tn5 is reflected in ChIL–seq, which use Tn5 to complete the transposition of ChIL probe that comprises the secondary antibody and ChIL DNA containing a T7 promoter and a primer sequence for the sequencing library preparation, including a mosaic end for Tn5 transposase binding.
Harada, A., Maehara, K., Handa, T., Arimura, Y., Nogami, J., Hayashi-Takanaka, Y., ... & Ohkawa, Y. (2019). A chromatin integration labelling method enables epigenomic profiling with lower input. Nature cell biology, 21(2), 287-296.
Handa, T., Harada, A., Maehara, K., Sato, S., Nakao, M., Goto, N., ... & Kimura, H. (2020). Chromatin integration labeling for mapping DNA-binding proteins and modifications with low input. Nature Protocols, 1-27.
Reviewer 3 Report
This manuscript reviewed recently developed genome DNA sequencing techniques (e.g., ATAC-Seq and LIANTI) using Tn5 transsposase mutagenesis. Perhaps this technology can overcome the limitations of modern next-generation sequencing (NGS) using short DNA length sequencing with the third long sequencing revolution, which contributes to the definition of an accurate entire genome DNA sequence. Therefore, this review topic is a great opportunity to explain the sequencing technology developed quickly in the field of genome research. Comparing different sequencing techniques for different objectives, such as Tn5 transposition mechanisms, epigenetics, 3D genome structure, genome variation, long fragment reading, and open chromatic structure, is a very interesting and effective task. This manuscript is suitable for publication after correct modification based on the following comments:
Line 20-21: TN5 must be changed to Tn5.
Lines 49-52: Authors must change the sentence correctly.
In general, many result figures are not correctly quoted. Need more information about the source data.
Line 110: remove gap from reference #23.
Line 112: Need to change sentence structure - ‘the difference in the magnification of different regions is significant different’ > ‘the difference in magnification by area varies greatly’
Line 153: delete comma before reference number 24.
Line 154: reference number 25 followed by 1 spacer required.
Line 174: Authors must change FARE-Seq to FAIRE-Seq, especially in the supplementary data legend.
Lines 193-194: deleted comma after each panel A, B, and C. and also Fig 6.
Lines 200-201: Peters et al~(#38) must be changed exactly to Wang et al (#42). Please refer to second generation seq and third generation seq for NGS technology.
In Fig 5 panel A, The DNA maker should be changed to a DNA marker.
Line 275: He’s lab~ the cited number is wrong. Need #51 change to #54.
Line 285: please use the term itChIP correctly and correctly anywhere in the text.
Line 287: what is the FRiP? Additional description is required.
Line 322: ‘these Tn5 transpose-involved or’ change to ‘these Tn5 transposase-involved or’
Line 334: Rubin et al~ reference number is 61
Line 337: Lu et al ~ not quoted. Need to correct quote an accurate paper.
Lines 362-382: For supplementary data, most of the original materials have been modified, so accurate quotes are required.
For reference, much of the information quoted does not follow the journal guideline for reference. In particular, the number 5 is a non-forming illustration. #37, 43, 44, 50 etc should not use capital letters in the title of the paper. #65 and 66 have to fill volume and page number. #65 is 21:137-150 and #66 is 578:455-460.
In Fig 3, panels A, B, and C are from Chen et al. (2017) paper Figure 1. Therefore, full citation and pictorial original references are required.
In Fig 5 panel A, the gel photo is from Amini et al (2014) reference #21. Panel C came from Wang et al modified fig 1.
Please carefully deal with supplementary data and the correct fully cited papers.
Suppl#1: from ref#23 Fig. 2c
Suppl#2: need to correct figure from ref #23 Fig. 4d
Suppl#4-5: from ref #45. Add a detailed description of the selected figure and data set. Ref#45 is wrong. #48 is correct from Lister et al Nature paper Fig. 1d and Fig. 3a.
Suppl #6: from ref. #54 Fig. 3A. The names PAT-T5 and PAT-T7 have changed to each other, so please correct them.
Suppl #7: from ref#55 Fig. 3a
Author Response
This manuscript reviewed recently developed genome DNA sequencing techniques (e.g., ATAC-Seq and LIANTI) using Tn5 transsposase mutagenesis. Perhaps this technology can overcome the limitations of modern next-generation sequencing (NGS) using short DNA length sequencing with the third long sequencing revolution, which contributes to the definition of an accurate entire genome DNA sequence. Therefore, this review topic is a great opportunity to explain the sequencing technology developed quickly in the field of genome research. Comparing different sequencing techniques for different objectives, such as Tn5 transposition mechanisms, epigenetics, 3D genome structure, genome variation, long fragment reading, and open chromatic structure, is a very interesting and effective task. This manuscript is suitable for publication after correct modification based on the following comments:
Response: Thank you very much for your comments and suggestions, which were of great help in improving our manuscript. We have revised the manuscript according to your comments and suggestions.
Line 20-21: TN5 must be changed to Tn5.
Response: Thank you very much for your suggestions. Done.
Lines 49-52: Authors must change the sentence correctly.
In general, many result figures are not correctly quoted. Need more information about the source data.
Response: Thank you very much for your suggestions. Done.
Line 110: remove gap from reference #23.
Response: Thank you very much for your suggestions. Done.
Line 112: Need to change sentence structure - ‘the difference in the magnification of different regions is significant different’ > ‘the difference in magnification by area varies greatly’
Response: Thank you very much for your suggestions. Done.
Line 153: delete comma before reference number 24.
Response: Thank you very much for your suggestions. Done.
Line 154: reference number 25 followed by 1 spacer required.
Response: Thank you very much for your suggestions. Done.
Line 174: Authors must change FARE-Seq to FAIRE-Seq, especially in the supplementary data legend.
Response: Thank you very much for your suggestions. Done.
Lines 193-194: deleted comma after each panel A, B, and C. and also Fig 6.
Response: Thank you very much for your suggestions. Done.
Lines 200-201: Peters et al~(#38) must be changed exactly to Wang et al (#42). Please refer to second generation seq and third generation seq for NGS technology.
Response: Thank you very much for your suggestions. Done.
In Fig 5 panel A, The DNA maker should be changed to a DNA marker.
Response: Thank you very much for your suggestions. Done.
Line 275: He’s lab~ the cited number is wrong. Need #51 change to #54.
Response: Thank you very much for your suggestions. Done.
Line 285: please use the term itChIP correctly and correctly anywhere in the text.
Response: Thank you very much for your suggestions. Done.
Line 287: what is the FRiP? Additional description is required.
Response: Thank you very much for your suggestions. FRiP is the abbreviation of the fraction of reads in peaks. We have added instructions of FRiP.
Line 322: ‘these Tn5 transpose-involved or’ change to ‘these Tn5 transposase-involved or’
Response: Thank you very much for your suggestions. Done.
Line 334: Rubin et al~ reference number is 61
Response: Thank you very much for your suggestions. Done.
Line 337: Lu et al ~ not quoted. Need to correct quote an accurate paper.
Response: Thank you very much for your suggestions. Done.
Lines 362-382: For supplementary data, most of the original materials have been modified, so accurate quotes are required.
Response: Thank you very much for your suggestions. Done.
For reference, much of the information quoted does not follow the journal guideline for reference. In particular, the number 5 is a non-forming illustration. #37, 43, 44, 50 etc should not use capital letters in the title of the paper. #65 and 66 have to fill volume and page number. #65 is 21:137-150 and #66 is 578:455-460.
Response: Thank you very much for your suggestions. Done.
In Fig 3, panels A, B, and C are from Chen et al. (2017) paper Figure 1. Therefore, full citation and pictorial original references are required.
Response: Thank you very much for your suggestions. Done.
In Fig 5 panel A, the gel photo is from Amini et al (2014) reference #21. Panel C came from Wang et al modified fig 1.
Response: Thank you very much for your suggestions. Done.
Please carefully deal with supplementary data and the correct fully cited papers.
Suppl#1: from ref#23 Fig. 2c
Response: Thank you very much for your suggestions. Done.
Suppl#2: need to correct figure from ref #23 Fig. 4d
Response: Thank you very much for your suggestions. Done.
Suppl#4-5: from ref #45. Add a detailed description of the selected figure and data set. Ref#45 is wrong. #48 is correct from Lister et al Nature paper Fig. 1d and Fig. 3a.
Response: Thank you very much for your suggestions. Done.
Suppl #6: from ref. #54 Fig. 3A. The names PAT-T5 and PAT-T7 have changed to each other, so please correct them.
Response: Thank you very much for your suggestions. Done.
Suppl #7: from ref#55 Fig. 3a
Response:Thank you very much for your suggestions. Done.
Round 2
Reviewer 1 Report
In the revised ms provided, the Authors only marginally addressed what I suggested in the previous round of revision.
The example introduced in the Introduction are jus t a mere list of studies where Tn5 has been used, but it lacks a clear perspective of the utility of the method since it is not clear what actually Tn5 is useful for.
The other suggestion about the evolutionary and taxonomic context of Tn5, although Authors stated they have introduced something, as not been addressed at all. A few words added do not give any idea of the diversity of Tn5 elements (if any), about the natural abundance among organisms of Tn5 (if any) and so on. They just added that it was found in Escherichia coli (mispelled in the ms, by the way). I suggest the reading of the followings, for more info: Berg et al., 1984. Mol Biol Evol 1:411-422; Biel and Hartl, 1983 Genetics 103:581-592. Probably more refrences relate on similar topics.
Overall, I am not satisfied with the revision provided. The ms still appears well written and well detailed but still looks like a technical report and, in my opinion, lacks of completeness.
Author Response
In the revised ms provided, the Authors only marginally addressed what I suggested in the previous round of revision.
The example introduced in the Introduction are jus t a mere list of studies where Tn5 has been used, but it lacks a clear perspective of the utility of the method since it is not clear what actually Tn5 is useful for.
The other suggestion about the evolutionary and taxonomic context of Tn5, although Authors stated they have introduced something, as not been addressed at all. A few words added do not give any idea of the diversity of Tn5 elements (if any), about the natural abundance among organisms of Tn5 (if any) and so on. They just added that it was found in Escherichia coli (mispelled in the ms, by the way). I suggest the reading of the followings, for more info: Berg et al., 1984. Mol Biol Evol 1:411-422; Biel and Hartl, 1983 Genetics 103:581-592. Probably more refrences relate on similar topics.
Overall, I am not satisfied with the revision provided. The ms still appears well written and well detailed but still looks like a technical report and, in my opinion, lacks of completeness.
Response:Thank you very much for your comments and suggestions.We added the background knowledge description of Tn5 in the first part. This manuscript aims to provide various applications of Tn5 in the process of chromatin research. There may be many shortcomings in the early background description, and your suggestions have made us realize this.